# Limited Additive Diagnostic Impact of Isolated Gastrointestinal Involvement for the Triage of Children with Suspected COVID-19

**DOI:** 10.3390/children9010041

**Published:** 2022-01-02

**Authors:** Désirée Caselli, Claudio Cafagno, Daniela Loconsole, Annamaria Giannini, Francesco Tansella, Annalisa Saracino, Maria Chironna, Maurizio Aricò

**Affiliations:** 1Pediatric Infectious Diseases, Giovanni XXIII Children Hospital, Azienda Ospedaliero Universitaria Consorziale Policlinico, 70124 Bari, Italy; desiree.caselli@policlinico.ba.it (D.C.); claudio.cafagno@hotmail.it (C.C.); aerius2@libero.it (A.G.); s.simonettis@policlinico.ba.it (F.T.); 2Hygiene Section, Department of Biomedical Sciences and Human Oncology, University of Bari, 70124 Bari, Italy; daniela.loconsole@uniba.it (D.L.); maria.chironna@uniba.it (M.C.); 3Clinic of Infectious Diseases, Department of Biomedical Sciences and Human Oncology, University of Bari, Azienda Ospedaliero Universitaria Consorziale Policlinico, 70124 Bari, Italy; annalisa.saracino@uniba.it; 4Strategic Control, Azienda Ospedaliero Universitaria Consorziale Policlinico, 70124 Bari, Italy

**Keywords:** COVID-19, patient triage, gastrointestinal symptoms

## Abstract

The strategy for the selection of patients with a suspected SARS-CoV-2 infection is relevant for the organization of a children’s hospital to provide optimal separation into COVID-19 and non-COVID-19 areas and pathways. We analyzed the proportion of children with COVID-19 presenting with gastrointestinal (GI) symptoms in 137 consecutive patients admitted between January 2020 and August 2021. GI symptoms were present as follows: diarrhea in 35 patients (26%), vomiting in 16 (12%), and both of them in five (3%); the combination of fever, respiratory symptoms, and diarrhea was observed in 16 patients (12%). Of the 676 adult patients with COVID-19 admitted to our hospital in the same time interval, 62 (9.2%) had diarrhea, 30 (4.4%) had vomiting, and 11 (1.6%) had nausea; only one patient, a 38-year-old male, presented with isolated GI symptoms at the diagnosis. Although diarrhea was observed in one quarter of cases, one-half of them had the complete triad of fever, respiratory syndrome, and diarrhea, and only five had isolated diarrhea, of which two were diagnosed with a Campylobacter infection. The occurrence of either respiratory symptoms or gastrointestinal symptoms in our patients was not related to the patient age, while younger children were more likely to have a fever. Of the 137 patients, 73 (53%) could be tested for their serum level of SARS-CoV-2 specific IgG antibodies. The observed titer ranged between 0 (n = 3) and 1729 BAU/mL (median, 425 BAU/mL). Of 137 consecutive patients with COVID-19 admitted to our referral children’s hospital, only three presented with an isolated GI manifestation. It is interesting to note that this finding turned out to be fully in keeping with what was observed on adult patients with COVID-19 in our hospital. The additive diagnostic impact of gastrointestinal involvement for the triage of children with suspected COVID-19 appears limited.

## 1. Introduction

Children have reduced severe acute respiratory syndrome coronavirus 2 (SARS-CoV-2) infection rates and a substantially lower risk for developing severe coronavirus disease 2019 (COVID-19) compared with adults [1,2]. In a recent study, Loske et al. provided evidence that the airway immune cells of children are primed for virus sensing, resulting in a stronger early innate antiviral response to SARS-CoV-2 infection than in adults [3]. Yet, the Delta Variant of Concern (VOC), formerly known as the Indian VOC or B 1.617.2, rapidly became the dominant strain of SARS-CoV-2 in most western countries, and its spread was found mainly in younger, more affluent groups [4]. The respiratory syndrome (fever, cough, respiratory difficulties) is the main feature and thus the main target for clinical screening of children with suspected COVID-19 [5,6,7,8,9,10]. Gastrointestinal (GI) symptoms related to COVID-19, such as vomiting, diarrhea, and abdominal pain, have emerged among extrapulmonary manifestations [11,12,13,14,15,16,17,18,19,20,21,22]. 

In a recent meta-analysis of nine studies comprising 280 children, Akobeng et al. reported that the pooled prevalence of gastrointestinal symptoms was 22.8%. The pooled prevalence of diarrhea, vomiting, and abdominal pain was, respectively, 12.4%, 10.3%, and 5.4% [23]. 

Robust evidence that COVID-19 may present in children with an isolated GI manifestation would have implications in the screening strategy and patient triage on admission to hospital and outpatient services. Apulia is a large region in Southern Italy with about four million inhabitants, including 652,754 children. During the so-called “first wave” of the pandemic, in our region, children were only slightly affected by the SARS-CoV-2 infection, both in terms of clinical severity and hospitalizations [24]. Yet, recently, the younger population has been increasingly involved, including children, although with a persistently limited disease burden. 

Here, we report our experience on presenting clinical manifestations and laboratory findings of children with confirmed SARS-CoV-2 infection, admitted to a tertiary care children hospital in southern Italy during the COVID-19 pandemic.

## 2. Materials and Methods

Setting. The “Giovanni XXIII” Children Hospital of Bari is a 156-bed teaching institution, serving as a referral for the entire region. By regional public health care strategy, all children with documented SARS-CoV-2 infection deserving admission are expected to be referred to our hospital. As of 25 August 2021, a total number of 3,192,523 diagnostic tests have identified 261,814 positive cases in the Apulia region [25]. 

To prevent children from undue nosocomial SARS-CoV-2 exposure, the hospital designed and implemented two specific pathways aimed at separating the inpatients since their admission, according to their SARS-CoV-2 infection status [25]. At the front desk, all patients are triaged: they are asked to report their social/familial risk of exposure to a person diagnosed with COVID-19 and, furthermore, the presence of clinical manifestations typical of COVID-19 in children (i.e., in particular, persistent fever > 38 °C with respiratory syndrome). If so, they are addressed to the COVID-pathway, consisting of a dedicated Fast Track for initial evaluation, aside but independent of the emergency room. Since 20 April 2020, all patients deserving admission to the ward are temporarily allocated in a “Grey Area” in which a nasopharyngeal swab is performed. Upon evidence of negativity at this real time-PCR test, the patient can be released to the competent ward based on the diagnosis of admission. Otherwise, all patients already known to be SARS-CoV-2 positive or resulting positive at the real time-PCR test performed on the nasopharyngeal swab screening collected in the hospital are admitted to the “Red Area”. Both the Grey and the Red areas are part of the Infectious Diseases ward of the children’s hospital, fully equipped with negative pressure isolation. This setting allowed us to completely separate patients negative for SARS-CoV-2 and thus to define a COVID-free hospital from those who are waiting for testing results and those who tested positive. 

Patients. Inclusion criteria: all consecutive patients with a confirmed SARS-CoV-2 infection admitted to the “Giovanni XXIII” Children Hospital of Bari for any cause from January 2020 to August 2021 were enrolled. Exclusion criteria were only lack of informed consent for diagnosis and treatment and age older than 18 years. Causes of hospitalization and symptoms at presentation were analyzed. Patients were screened for SARS-CoV-2 infection on admission to prevent intra-hospital spread of the virus. In the early follow-up, patients were offered detection of specific anti-SARS-CoV-2 antibodies. Clinical signs and symptoms were reported based on the findings at clinical evaluation in our hospital. 

Data on the presenting features of adult patients with COVID-19 admitted to our adult hospital were also collected for comparison. 

Diagnostic procedures. The diagnosis of SARS-CoV-2 infection was established based only on the results of the real time-PCR test on nasopharyngeal swab. All samples were analyzed at the Laboratory of Molecular Epidemiology and Public Health of the Hygiene Unit (A.O.U.C. Policlinico Bari), which is the coordinating center of the Regional Laboratory Network for SARS-CoV-2 diagnosis. The RNA was extracted using the MagMAX Viral/Pathogen Nucleic Acid Isolation kit (Thermo Fisher Scientific, Waltham, MA, USA). The molecular test was performed using a three-target (N, ORF1ab, and S genes) commercial multiplex real-time PCR assay from Thermo Fisher Scientific (TaqPath RT-PCR COVID-19 Assay). The results were available to the attending clinician within 6 h. 

All serum samples collected in the subset of patients were tested for anti-SARS-CoV-2 IgG antibodies. Detection of anti-spike IgG (anti-S1) was performed through an ELISA method (Euroimmun, Lubeck, Germany). 

Data analysis. Anonymized data were stored in a Microsoft Office Access^®^ database and were synthesized in terms of mean ± sd or median [min, max] and frequencies with relative percentages, depending on the variables’ nature. A logistic regression model was applied in order to assess the association between age and outcomes of interest. The results were expressed with an Odds Ratio (OR) and the 95% confidence interval (CI). A *p*-value < 0.05 was considered statistically significant. All the analyses were carried out with SAS software, release 9.4. 

Ethical considerations. Informed consent was obtained from the patients’ parents or legal guardian in all cases. This observational study (n. 6359) was approved by the institutional IRB on 29 April 2020.

## 3. Results

Presenting features. Between February 2020 and August 2021, a total of 137 children and adolescents with a SARS-CoV-2 infection were admitted to our children hospital. There were 73 males and 64 females. Their age at diagnosis ranged between 5 days and 17.9 years (median, 32 months). 

Fifteen patients (11%) were unexpectedly found to be positive for SARS-CoV-2 upon screening preceding admission for one of the following conditions, with all considered to be not related to COVID-19: neuro-psychological disorders (n = 4), trauma (n = 3), bone pain (n = 2), cancer (n = 2), pancreatitis, ovary mass, hematuria, and isolated anemia (one each). 

In the remaining 122 patients (89%), fever was present in 98 (81%), respiratory symptoms (cough, breathing difficulties) in 78 (64%), and the combination of fever and respiratory symptoms was present in 64 patients (47%). 

GI symptoms were present as follows: diarrhea in 35 patients (25%), vomiting in 16 (13%), and both of them in 5 (4%); isolated diarrhea was observed in 5 patients (4%)—in two of these, Campylobacter enteritis was then documented. The combination of fever, respiratory symptoms, and diarrhea was observed in 16 patients (12%). The distribution of the presenting features is summarized in Table 1.

The occurrence of respiratory symptoms was not related to the age of the patients (OR point estimate 0.97, C.I. 0.92–1.02; *p*-value 0.254) nor was it related to the occurrence of gastrointestinal symptoms (OR point estimate 0.98, C.I. 0.92–1.04; *p*-value 0.472). Otherwise, younger children were more likely to have a fever (OR point estimate 0.93, C.I. 0.87–0.95; *p*-value 0.013).

Of the 676 adult patients with COVID-19 admitted to our hospital in the same time interval, 62 (9.2%) had diarrhea, 30 (4.4%) had vomiting, and 11 (1.6%) had nausea; only one patient, a 38-year-old male, presented with isolated GI symptoms at the diagnosis. 

Detection of Serum SARS-CoV-2 Specific Antibodies

Another topic we wanted to explore in our pediatric cohort was the pattern of persistence of serum antibodies specific for SARS-CoV-2 at a short-term follow-up. Since all patients did not accept to come back to the hospital for a follow-up visit, only 73 of the 137 patients (53%) could be tested. The serum level of SARS-CoV-2 specific IgG antibody titers ranged between 0 and 1729 BAU/mL (median, 425 BAU/mL). Three patients resulted negative for IgG anti-S. The distribution of the SARS-CoV-2 specific serum IgG antibody titers according to the time elapsed from the diagnosis of COVID-19 is shown in Figure 1.

## 4. Discussion

Here, we report the presenting features of 137 consecutive children and adolescents admitted to our children’s hospital in whom a SARS-CoV-2 infection was confirmed by reverse transcription-polymerase chain reaction (RT-PCR) on nasopharyngeal swabs. They were diagnosed during the period 21 January 2020 to 9 August 2021.

In order to reduce the risk of nosocomial spread of SARS-CoV-2, at the front desk we triage all children for familial exposure to SARS-CoV-2 and for the presence of a respiratory syndrome (fever, cough, or shortness of breath). Patients reporting familial exposure and/or respiratory syndrome are considered as potential cases of COVID-19 and thus evaluated in a dedicated, fast-track pathway [26].

In our series, 11% of patients had been admitted because of reasons considered to be unrelated to COVID-19, including cancer, trauma, or programmed surgery. Yet, their mandatory pre-admission screening revealed RT-PCR positivity; thus, their diagnosis of a SARS-CoV-2 infection was considered as “occasional”.

The remaining 122 patients had symptoms considered compatible with COVID-19. The vast majority of patients presented with fever, observed in more than 80%; respiratory syndrome was observed in almost 60% of the patients. This is in keeping with reports from other geographic areas [27,28]. The presence of respiratory or gastrointestinal symptoms was not related to the patient age, while younger children were more likely to have fever.

In a recent report of the Cochrane COVID-19 Diagnostic Test Accuracy Group, signs and symptoms were analyzed to determine if a patient presenting in primary care or hospital outpatient settings likely had COVID-19. However, still none of the studies presented any data on children separately [29]. Clinical characteristics of patients with COVID-19 presenting with GI symptoms as initial symptoms were retrospectively analyzed by Yang et al., but all patients included were 18 years old and above [30]. In order to understand if GI symptoms are worth incorporating and thus drive the initial triage process, we calculated the proportion of patients presenting with GI symptoms in the absence of a respiratory syndrome. Although diarrhea was observed in one quarter of cases, one-half of them had the complete triad of fever, respiratory syndrome, and diarrhea. Only five patients had isolated diarrhea at presentation, and in two of them, a Campylobacter infection was diagnosed. Thus, out of 137 patients, only three, i.e., 2%, presented with isolated diarrhea not justified by another associated condition. It is interesting to note that this finding turned out to be fully in keeping with what was observed on adult patients with COVID-19 in our hospital, of which only 9% had diarrhea, and only one presented with isolated GI symptoms at the diagnosis.

The proportion of patients with GI involvement observed in our experience is also comparable to the findings from other geographic areas [17,18,27,31]. For instance, of 244 consecutive children with COVID-19 from Wuhan during the period 21 January to 20 March 2020, 34 (13.9%) presented with GI symptoms (one of the following: diarrhea; nausea, and vomiting; abdominal pain; decreased feeding) on admission [15]. Remarkably, all 34 also had signs or symptoms of respiratory syndrome; seven had an acute upper respiratory infection, 25 mild pneumonia, and 2 critical pneumonia; fever was present in 24 patients (71%).

Another topic we wanted to explore in our pediatric cohort was the pattern of persistence of serum antibodies specific for SARS-CoV-2. Immune response to SARS-CoV-2 has been the object of an impressive number of studies within the last year [3,32,33,34]. A rapid decay of anti-SARS-CoV-2 IgG has been reported [35,36]. Different to adults, antibody profiles in children were not related to disease severity, suggesting distinct primary SARS-CoV-2 immune responses in children and adults [36]. Children have been reported to show a SARS-CoV-2-specific antibody response limited to IgG anti-S antibodies with a low level of neutralizing activity compared to adult COVID-19 cohorts. Denina et al. reported that 20 of 24 (83%) children back for a medical evaluation, on average 35 days post-discharge, had detectable levels of IgG directed toward SARS-CoV-2 using an ELISA assay (In3diagnostic Eradikit COVID-19, Turin, Italy) [37]. Waterfield et al. reported a group of 68 children with a SARS-CoV-2 infection in whom the presence of antibodies and the mean antibody titer was not influenced by age [38]. To address this issue, we proposed a follow-up visit to all patients, but not all of the patients accepted, confirming the very benign outcome of the disease. Thus, only 73 of the 137 patients (53%) could be tested. Overall, at a median time of 3.5 months from the diagnosis, persistence of SARS-CoV-2 specific antibodies was observed in all but three tested patients. This finding is in line with previous studies since it has been suggested that SARS-CoV-2 specific IgG response is sustained for some months after primary infection and declines thereafter [38,39]. A SARS-CoV-2 infection produces both humoral and cell-mediated responses [40] and results in the generation of neutralizing antibodies [41]. However, conventional tests for SARS-CoV-2-specific antibodies do not allow for the determination of whether a protective immune response is developed. Moreover, a correlate of protection based on the value of SARS-CoV-2 IgG is not available yet. On the other side, a growing body of evidence confirms that protection against SARS-CoV-2 is not dependent on antibody response only, while, as expected, the T-cell response plays a major role [42,43], although this is much more difficult to document in individual patients.

This study has limitations. The number of patients evaluated for their presenting features is not extremely high, although they have been seen in the same hospital, and thus, data collection and evaluation were uniform and complete. Thus, the limited impact of isolated diarrhea that we documented deserves independent confirmation on other comparable cohorts. Furthermore, monitoring of serum antibody titers was not performed prospectively on individual patients. Not surprisingly, since the vast majority of the children had a mild disease, the families were not keen to bring their children for repeated follow-up visits [6,19,37]. Thus, we describe the pattern of the titer of SARS-CoV-2 specific IgG antibodies based on single determinations, which related to the time elapsed from clinical manifestation of the disease. Yet, since there is no evidence of reason to suspect that the curve of decline of antibody titers are multi-phasic, the picture described appears reliable.

## 5. Conclusions

In conclusion, we consider the combination of fever and respiratory symptoms still the most appropriate for patient triaging in our hospital setting. Although diarrhea may be frequently associated at the disease onset, isolated diarrhea rather appears not to be considered as a robust individual marker of possible COVID-19 in children. A serum antibody titer, although of interest for assessment of a previous possibly undiagnosed infection, does not allow, so far, for predicting the persistence of specific immunity. While vaccination of all adults and adolescents is a clear and absolute must for public health strategies, vaccination for SARS-CoV-2 in younger children still awaits data on safety and efficacy, which we hope can become available in the near future [44,45].

## Figures and Tables

**Figure 1 children-09-00041-f001:**
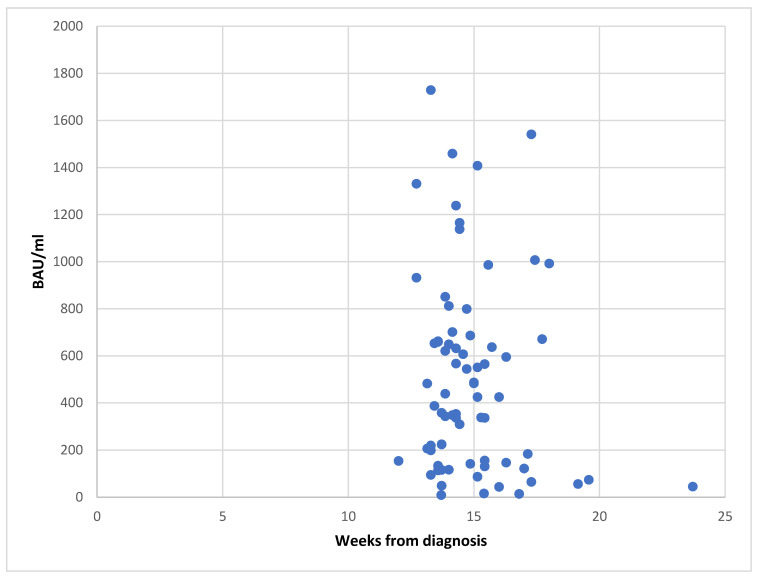
Distribution of the serum level of SARS-CoV-2 specific IgG antibodies by time from diagnosis in 73 children with COVID-19. At a median time of 3.5 months from the diagnosis, persistence of SARS-CoV-2 specific antibodies was observed in all but three tested patients, although with a widely variable range of titers.

**Table 1 children-09-00041-t001:** Main clinical signs or symptoms of children admitted at the “Giovanni XXIII” Children Hospital of Bari between January 2020 and August 2021 and diagnosed with COVID-19.

**Total Patients**	137
Gender	
Female	64
Male	73
Age	
Range (years)	0–17.9
Median	2.6
Patients with at least one COVID-19 related sign or symptom (n = 122)	
Type	N. (%)
Fever	98 (81%)
Respiratory symptoms (cough, breathing difficulties)	79 (58%)
Fever and respiratory symptoms	64 (47%)
Diarrhea	35 (25%)
Isolated diarrhea	5 (3.6%)
Vomiting	16 (13%)
Vomiting and diarrhea	5 (4%)
Abdominal pain	1 (0.7%)
Fever, respiratory symptoms, and diarrhea	16 (12%)
Patients with Non-COVID-19 related signs or symptoms (n = 15)	
Neuro-psychological disorders	4 (3%)
Trauma	3 (2%)
Bone pain	3 (2%)
Cancer	2 (1%)
Pancreatitis	1 (0.7%)
Ovary mass	1 (0.7%)
Hematuria	1 (0.7%)

## Data Availability

The data presented in this study are available on request from the corresponding author.

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
