# Peer review of "Limited Additive Diagnostic Impact of Isolated Gastrointestinal Involvement for the Triage of Children with Suspected COVID-19"

_children, 2022, doi:10.3390/children9010041_

Round 1

Reviewer 1 Report

The authors have addressed the recommendations from the referee

Author Response

Rev.. The authors have addressed the recommendations from the referee

Re.: No action is required.

Reviewer 2 Report

In this study, Caselli D et al. analyze the impact of the isolated gastrointestinal involvement for the triage of children with suspected COVID- 19, which was found to be limited.

Although interesting, the study is somewhat biased by the methods employed which do not allow the reader to understand the real impact of these findings in the routine daily practice.

First, the control group should be determined with children without Sars-CoV-2 infection presenting in the same period to the Emergency Department, since gastrointestinal symptoms incidence was low in all the pandemic period due to the low spread of gastroenteritis and other infections in general.

Second, authors do not report the incidence of children presenting after a contact with a Sars-CoV-2 positive individual: parents with children with such condition may overestimate the impact of respiratory symptoms and underestimate gastrointestinal ones.

Third, it is not clear the comparison with adults, which display very different features of COVID-19.

Fourth, authors analyzed concomitant gastrointestinal infections (such as Campylobacter) but they did not analyze concomitant respiratory infections. This may bias the results section. What was the incidence of MISC with gastrointestinal involvement?

Fifth, the analysis of the antibody response seems to have limited value, since not all patients were reevaluated and they were not followed over time.

In conclusion, authors reached too far with a limited sample population, with the statement of the limited value of considering gastrointestinal symptoms at ED admission. This is in contrast with other previous reports which highlight these features of COVID-19 (https://www.medrxiv.org/content/10.1101/2020.12.29.20248994v1.full, https://www.ncbi.nlm.nih.gov/pmc/articles/PMC7324269/, https://www.ncbi.nlm.nih.gov/pmc/articles/PMC7095065/).

In particular:

Introduction: slightly repetitive in the first part In the second part, authors should focus on the analysis of the isolated gastrointestinal manifestation in the current literature (https://pubmed.ncbi.nlm.nih.gov/34249319/)

Methods: well presented. Authors should add inclusion and exclusion criteria, since they are crucial for the reader. 

Results: should be extensively revised. "Respiratory symptoms" is too indistinct. Parents are known to overestimate or underestimate this symptoms depending on the situations. Did the reported symptoms were confirmed by clinical signs detection by the physician?

Conclusion: Also in the light of the recent experience of unexpected TTP-like complication associated with adenovirus-associated vaccines (43), the decision to make active immunization for SARS-CoV-2 mandatory in low-risk categories, including children, appears as a step to be taken with caution. Honestly, I cannot understand which part of your study support this statement.

Author Response

Rev.: 

  1. First, the control group should be determined with children without Sars-CoV-2 infection presenting in the same period to the Emergency Department, since gastrointestinal symptoms incidence was low in all the pandemic period due to the low spread of gastroenteritis and other infections in general.

Re.: The issue could be of interest but unfortunately this feature fall outside the study design. Thus, this control cohort is not available.

2. Second, authors do not report the incidence of children presenting after a contact with a Sars-CoV-2 positive individual: parents with children with such condition may overestimate the impact of respiratory symptoms and underestimate gastrointestinal ones.

Re.: The information on familial exposure was only used for triaging the patients at the ER. The reported presenting features are not affecetd by this information and only comprise what is observed by the physician. Thus, this possible bias can be safely excluded.  

3. Third, it is not clear the comparison with adults, which display very different features of COVID-19.

Re.: The comparison with adults was made in the attempt to have a perspective on the clinical patterns in our geographic area, potentially related to circulation of viral variants. 

4. Fourth, authors analyzed concomitant gastrointestinal infections (such as Campylobacter) but they did not analyze concomitant respiratory infections. This may bias the results section. What was the incidence of MISC with gastrointestinal involvement?

Re.: as stated, the symptoms reported deal with the presenting feature. None of the children presented with a MIS-C picture. The follow-up of those patients was not part of the study design. 

5. Fifth, the analysis of the antibody response seems to have limited value, since not all patients were reevaluated and they were not followed over time.

Re.: we acknowledge that the contribution of data on serology in patients with SARS-CoV-2 infection now turns out to be quite limited, in general, and also in children. Yet, at the time we design this study, we considered that data on serology could have been of interest, in terms of variablility of the titer and duration of the responsa. 

In conclusion, authors reached too far with a limited sample population, with the statement of the limited value of considering gastrointestinal symptoms at ED admission. This is in contrast with other previous reports which highlight these features of COVID-19 (https://www.medrxiv.org/content/10.1101/2020.12.29.20248994v1.full, https://www.ncbi.nlm.nih.gov/pmc/articles/PMC7324269/, https://www.ncbi.nlm.nih.gov/pmc/articles/PMC7095065/).

In particular:

6. Introduction: slightly repetitive in the first part In the second part, authors should focus on the analysis of the isolated gastrointestinal manifestation in the current literature (https://pubmed.ncbi.nlm.nih.gov/34249319/)

Re.: The firts part was shortened and the second part modified. The proposed reference was taken into consideration and cited. 

7. Methods: well presented. Authors should add inclusion and exclusion criteria, since they are crucial for the reader. 

Re.: This was done. 

8. Results: should be extensively revised. "Respiratory symptoms" is too indistinct. Parents are known to overestimate or underestimate this symptoms depending on the situations. Did the reported symptoms were confirmed by clinical signs detection by the physician?

Re.: Thank you for helping to clarify a potential pitfall. Maybe we had not specified enough that signs and symptomd reported are only those observed and documneted by our physician. None of them was mentioned only based on parents' reporting. This was openly mentioned. 

9. Conclusion: Also in the light of the recent experience of unexpected TTP-like complication associated with adenovirus-associated vaccines (43), the decision to make active immunization for SARS-CoV-2 mandatory in low-risk categories, including children, appears as a step to be taken with caution. Honestly, I cannot understand which part of your study support this statement.

Re. We acknowledge that this statement is not directly related to our findings. Thus, the statement has been deleted.  

Round 2

Reviewer 2 Report

This manuscript is improved and authors have addressed my suggestions.

One specific questions about the table:

- Is it possible to have a logistic regression of distribution of symptoms by age?This could be of great value, especially for younger children, in order to identify possible symptoms cluster by age.

Author Response

As suggested by the reviewer, regression analysis was performed to explore the association between age and the presence of main symptoms. The results have been introduced in the abstract, in the results and briefly commented in the discussion. 

This manuscript is a resubmission of an earlier submission. The following is a list of the peer review reports and author responses from that submission.

Round 1

Reviewer 1 Report

The authors present an interesting analysis of a series of cases of covid-19 in adolescents and infants in a hospital in Italy. The work is generally well done, but there are a few things to consider before proceeding with the publishing process.
Line 24 add "and August 2021".
Lines 127-129. please remove the text from these lines.
Line 241 provides a reference

Table 1 organize the content presented, the sociodemographic characteristics of the study population, symptoms, clinical presentation and IgG titers, as well as p values.

Figure 1 is not self-explanatory. 

It is necessary to add a section on ethical considerations and the number of the committee's approval act.

Reviewer 2 Report

This is a well written manuscript reporting the experience of the authors on presenting clinical manifestations and laboratory findings of children with confirmed SARS-CoV-2 infection, admitted to a tertiary care children hospital in southern Italy during the COVID-19 pandemic. The conclusions are drawn by a small cohort of patients and they do not offer much to the existing literature as there results have been described in several other studies from other parts of the world.